# All Optical Speckle Contrast-Based Vibration Sensor for Photoacoustic Signal Detection

**DOI:** 10.3390/s22093250

**Published:** 2022-04-23

**Authors:** Matan Benyamin, Zeev Zalevsky

**Affiliations:** Faculty of Engineering and the Nanotechnology Center, Bar Ilan University, Ramat Gan 5290002, Israel; zeev.zalevsky@biu.ac.il

**Keywords:** non-contact vibration detection, all-optical, photoacoustic, speckle contrast

## Abstract

Remote detection of photoacoustic signals is a well desired ability, enabling to perform advanced imaging in scenarios where contact is not possible. Various unique solutions have been suggested, including a camera-based speckle contrast photoacoustic detection. In this manuscript, a significant upgrade to the camera-based speckle contrast approach is presented and experimentally demonstrated. This solution is based on all-optical vibration sensing setup. The technique is based on spectral estimation of speckle pattern contrast and relies on several pre-developed works. First, it relies on the suggested application of speckle contrast to vibration sensing, and then on the realization of intensity pattern spectral manipulation, using a shearing interferometer. The method is evaluated and compared to traditional contrast estimation, and demonstrated in several applications in various vibration frequency band such as photoacoustic signal analysis and phonocardiographic heart sounds. The method is also applicable to measuring contrast changes due to a general speckle changing behavior, rather than surface vibration alone.

## 1. Introduction

The photoacoustic (PA) effect is a physical phenomenon in which optical energy is translated into an acoustical-mechanical response. When a short time optical energy (~ns temporal pulse width) is absorbed by a material and generates heat, the material may expand and initiate a mechanical wave. The equation governing this effect, under the assumptions of stress and thermal confinement, non-saturated absorption, and linear thermal expansion, is the photoacoustic wave equation [1]: (1)$$\left(\nabla^{2}-\frac{1}{v^{2}}\frac{\partial^{2}}{\partial t^{2}}\right)p\left(r,t\right)=\frac{\partial }{\partial t}\Gamma \mu_{a}\phi \left(r,t\right)$$
where v is the speed of sound, p is the resulted pressure, Γ is the unitless Grüneisen coefficient related to energy transfer efficiency, $\mu_{a}$ is the absorption coefficient and ϕ(r,t) is the fluence of the excitation source. Detection of PA signals, as other acoustical vibration signals, mostly requires direct contact with the vibrating object, and is based on using piezo electric transducers, in addition to impedance matching layer to avoid reflections and signal losses. However, in many applications direct contact is not favorable, or even not possible at all (i.e., brain surgeries [2], or ophthalmology [3]).

Non-contact optical methods for detecting vibration signals have been suggested. Most are based on either interferometry [4,5] or refractometry [6,7]. Those approaches require perfect specular reflection from the vibrating surface and complicated signal analysis, and in addition are strongly affected by external noise. In addition to minimal robustness, interferometric methods sometimes involve specific calibration that may be required rather often, thus making interferometry an unstable basis for real life scenarios of vibration signal detection.

Speckle-based systems have been offered as an alternative in various variants for detecting μm scale vibrations [8,9,10]. Those methods commonly rely on speckle generation—a coherent light source illuminates the sample, and the reflection from the sample’s rough surface cause a random phase distribution, which in turn results in a speckle pattern. This pattern demonstrates great yet adjustable sensitivity to subtle changes in the properties of the reflecting surface while using simple optical hardware. This sensitivity can be adjusted through various parameters in the setup—focal length, sample position relative to focal plane, etc.

Since imaging devices often present limited frame rate, not sufficient to explore the MHz bandwidth characterizing PA signals, Speckle contrast-based methods were suggested as an appropriate solution [11,12,13].

Speckles reflected from a sample will vary in high frequencies, during the acquisition time of the camera capturing the speckles. Since the camera averages over the exposure time in each frame, the frame will see, blurry when there is a variation in the speckle pattern thorough out the exposure time. More specifically, the faster and larger the variation in speckle patter, the blurrier the frame will get. Theoretically, the relation between the contrast, K the correlation function characterizing the intensity fluctuations of the speckle pattern, C(t), and the camera exposure time *T* can be described [14]:(2)$$K^{2}\left(T\right)=\frac{1}{T}\int_{0}^{T}C^{2}\left(t\right)dt$$

Empirically, contrast $K_{e}$ can be estimated by calculating:(3)$$K_{e}=\frac{\sigma \left(x\right)}{<I\left(x\right)>}$$
where σ(x) is the standard deviation and <I(x)> is the average of the pixel values in the frame.

Speckle contrast has been vastly used for flow imaging [15] and other applications. More recently, it was purposed as a technique for vibration signals detection, specifically applied to photoacoustic signals [13] but can trivially be extended to other vibration signals. Due to its integrative characteristic, contrast lacks the ability to differentiate between two different intra-frame behaviors that share the same integral value. For instance, continuous vibration in frequencies with period much shorter than the integration time of the camera, will result in the same contrast value, as long as they share amplitude. This may be a drawback of contrast-based methods. Benyamin presented a scheme containing temporal scanning in order to address this issue, and applied the idea to perform back projection-based image reconstruction, which uses temporal information of the vibration signal [12]. Even though the method used by Benyamin presents somewhat different signals, with limited temporal resolution compared to piezoelectric transducers, it was demonstrated how those signals do employ the required information in order to perform back-projection imaging.

The method presented by Benyamin requires a camera, a laser, and several simple optical components. The imaging scheme does require a minimal number of pixels, since it relies on statistical measurements—standard deviation and mean. Using a camera as an integral part of the detection setup derives some limitations and disadvantages, as detailed below, and indirectly it also derives a minimal integration time of the detector to allow sufficient energy to overcome noise.

In this work, we focus on applying the speckle contrast-based detection technique, along with its development already presented by Benyamin, but instead of using a camera, implementing the calculation by means of optical elements using only a simple single pixel detector. In order to do so, we replace the previously discussed statistical contrast calculation with a frequency-based one, $K_{f}$ which is more easily implemented via optical elements. The denominator of Equation (3), <I(x)>, is in fact the DC value on the spatial spectrum of the speckle pattern intensity. The information on intensity’s variance, which is originally formalized in the standard deviation operator, can be estimated by taking the higher frequencies of the spatial spectrum:(4)$$K_{f}=<S_{High}>-<S_{Low}>$$
where $<S_{High}>$ is the average of high spatial frequency energy of the speckle pattern, and $<S_{Low}>$ is the average of low frequency energy. The division operation was replaced with subtracting, which is easily implemented optically when using coherent light source. However, it may introduce noise and sensitivity to background illumination.

This calculation relies on the spatial frequency distribution of the speckle pattern, and therefore may be optically realized using the tools of Fourier optics. We would like to calculate the Fourier transform, or any other spatial spectral representation of the speckle pattern, and then to separate the high frequency band from the lower one. Then, an optical subtraction of those two is performed, provided the two channels—high and low frequency, are coherent.

Using a single pixel, instead of a camera contain millions of pixels, can be advantageous in various manners. In terms of acquisition time, single pixel detectors require significantly shorter acquisition time than high resolution cameras. This directly affects the possible frame rate and temporal resolution. Second advantage is computation time—camera-based setup requires calculation of speckle contrast performed on the image, after the acquisition. The single pixel setup presented here realizes this processing with optical elements, in the speed of light. Third, single pixel detectors do offer improved performance compared to cameras, in terms of dark noise, cost efficiency, and even extended spectral range [16,17,18].

In order to optically implement a spectral transformation to the speckle pattern intensity, excluding the phase, we use a method previously suggested by Mendelovic [19]. In this method, a rotating diffuser and a shearing interferometer including a beam splitter, corner prism, and a tilted mirror were used, realizing a cosine transform. The optical setup is described below. Using this setup results in the output plane that is recorded by the camera. The probe beam hit the vibrating sample, and a speckle pattern is reflected due to the surface roughness of the sample. First, the field passes through a rotating diffuser:(5)$$E_{1}\left(x_{1},t\right)=E\left(x,t\right)e^{i\phi \left(x_{1},t\right)}$$
where $E_{1}$ is the field on $x_{1}$ plane, and ϕ is the phase originating the rotating diffuser which aims to interfere with the coherence of the field. Then, the shearing interferometer containing a corner prism and a beam splitter:(6)$$\int_{-\infty }^{\infty }E_{1}\left(x_{1},t\right)e^{\left(\frac{\pi i}{\lambda z}x_{1}^{1}2+{x_{2}}^{2}\right)}dx_{1}2\cos\left(\frac{2\pi }{\lambda z}x_{1}x_{2}\right)dx_{1}\;E\left(x_{2},t\right)=\int_{-\infty }^{\infty }E_{1}\left(x_{1},t\right)e^{\frac{\pi i}{\lambda z}{\left(x_{1}-x_{2}\right)}^{2}}dx_{1}+\;\int_{-\infty }^{\infty }E_{1}\left(x_{1},t\right)e^{\frac{\pi i}{\lambda z}{\left(-x_{1}-x_{2}\right)}^{2}}dx_{1}=\;\int_{-\infty }^{\infty }E_{1}\left(x_{1},t\right)e^{\left(\frac{\pi i}{\lambda z}{x_{1}}^{2}+{x_{2}}^{2}\right)}dx_{1}2\cos\left(\frac{2\pi }{\lambda z}x_{1}x_{2}\right)dx_{1}$$
where λ is the light wavelength and z is the optical axis. By this point we may see that the light in $x_{2}$ is a cosine transform of the speckle pattern in $x_{1}$. In order to perform the spectral contrast estimation, we only need to subtract the central part of $x_{2}$, the lower frequencies, from its outer part. This is performed using a small wave plate, $g\left(x_{2}\right)$ which delays by a phase of π only the low frequency part of the spectrum:(7)$$I\left(x_{3}\right)=2g^{2}\left(x_{2}\right)\int_{-\infty }^{\infty }{\left|E\left(x_{1}\right)\right|}^{2}\cos^{2}\left(\frac{2\pi }{\lambda z}x_{1}x_{2}\right)dx_{1}$$

$I\left(x_{3}\right)$ is a cosine transform of the input plane ${\left|E\left(x_{1}\right)\right|}^{2}$, which is, in our case, the intensity of the speckle pattern, multiplied by a squared filter function $g\left(x_{2}\right)$ which subtracts the lower frequencies from the higher ones.

Despite having some drawbacks, such as increased sensitivity to overall background illumination, the spectral estimation of contrast can inherently overcome some challenges not addressed directly by the statistical measurement. For instance, if the probe illumination from which the speckle pattern is generated, is not spatially flat, and does exhibit some spatial behavior, Gaussian for example—the statistical estimation will aggerate both contrast due to speckle and contrast of the overall Gaussian shape of the beam, and the contrast of the speckle itself, which is the objective of the measurement, may be shadowed by the beam distribution contrast. This issue may be resolved digitally with sophisticated background subtraction, if a full image is taken. However, in the optical method presented here, one can smartly design the denominator of Equation (3) to be not only the DC term, but a band of low frequencies. This ways, Gaussian shape of the beam which is characterized by low spatial frequencies will be compensated seamlessly. This benefit is presented in the results.

Another benefit of the optical implementation is related to the coherence requirement from the illumination. In general, contrast-based measurement does require a relatively high contrast to begin with, which in turn translates into a high coherence requirement from the probe laser. One of the factors affecting this requirement is the minimal camera integration time determined by the used camera. The probe laser should be coherent in the time scale of that minimal integration time. However here we do not use a camera but rather a fast photodetector, allowing, in theory a much shorter integration time and therefor much less coherent illumination

This work establishes the all-optical setup for detecting time-resolved photoacoustic signals. The experimental demonstration realizes a single-point photoacoustic detection. However, the transition from single point signal to full photoacoustic amplitude imaging, as well as vibration imaging, can be done in two ways; either by mechanical raster scanning [13], or by back projection reconstruction [12].

In addition to photoacoustic signals, speckle contrast may also be used in order to quantify other phenomena—either vibration in various frequency bands, or even speckle changing not due to vibration but due to less organized scattering behavior. For instance, speckle reflected back from a blood vessel, change its contrast when blood flow and volume vary. This kind of usage has been applied in order to quantify blood pulsation [20]. A similar application, this time replacing the digital computation of speckle contrast by the optical one suggested here, is presented.

Another application presented here, containing vibrations of significantly lower frequencies, is chest vibration due to heartbeat, also known as phonocardiography [21]. In this scenario the probe beam is directed onto a human chest, instead of pumping a photoacoustic sample. The probe procedure was performed in a similar way to all other applications presented in this work.

## 2. Materials and Methods

The setup described in this manuscript realizes the following flow: (a) A speckle pattern is generated from the reflection of a probe beam; (b) phase distribution of the speckle pattern is interfered by a rotting diffuser; (c) a non-coherent cosine transform is performed on the speckle intensity; (d) the frequency image passes through a wave plate which causes the lower frequency area to retard by π; (e) the beam is focused and a difference between the higher and lower frequencies is received due to the π phase between them. The setup itself is described in Figure 1.

The pump laser used for the photoacoustic signal generation was a 1047 nm pulsed laser (pulse width 0.5 ns, Lightwave Electronics). The excitation spot was focused to a size of 500 μm using a 75 mm lens (THORLABS). The probe is a 2.5 mW CW 780 nm collimated laser diode (THORLABS, CPS780S) with a preset focusing lens. The sample for the photoacoustic signal reconstruction phase was thin-walled ink filled silicone tube, with a width of 50 μm.

First, the spectral approach of estimating contrast was tested and compared to the traditional statistical calculation (noted in Equation (3)). A speckle image was generated, and the contrast was gradually and synthetically reduced by smoothing the image with a moving average filter of different kernel sizes. The statistical calculation was then compared to the spectral one. The aim of the test was to support the usage of spectral-based contrast estimation. The results of this experiment are described in Figure 2, and referred to in the results below. This test can be also used as an SNR approximation of the all-optical spectral contrast estimation, compared to the traditional statistical estimation received from a camera setup. A significant reduction of the system’s output, as a result of vibration and contrast reduction resembles a good SNR of the system.

Since the measurement protocol of the detection itself is similar in both all-optical and camera based setups, the temporal resolution is exactly the same.

The signal captured by the detector in the purposed setup was shown to have a relation to the speckle pattern intensity image, which is captured by a camera in a traditional speckle contrast setup [13]. Then, in order to compare the two signals, statistical contrast with a camera and spectral contrast with the purposed optical setup, we may take a speckle image from a camera and synthetically transform it into what we would have received using the purposed all optical setup.

In order to demonstrate the ability of the technique in reconstructing vibration signal of photoacoustic origin, as well as other sources, the idea of spectral speckle contrast estimation is applied here to speckle patterns recorded in various scenarios. First, it is demonstrated on the detection of photoacoustic signals, using the setup described above.

To examine future applications of this unique vibration sensor, the same idea was applied on speckle patterns changing due to other physical phenomena: scattering due to red blood cells movement and chest movement due to beating heart, simply by taking measurements recorded without pump laser and replacing the photoacoustic sample by a human forehead [20] or a human chest, accordingly.

## 3. Results

An evaluation of the spectral contrast estimation approach, relative to the traditional statistical one, as detailed above is presented in Figure 2.

In Figure 3, the effect of a non-uniform beam shape is explored. It can be seen that the contrast degradation, relative to the initial speckle contrast, is minimal in the statistical estimation approach, whereas the spectral approach, which can be designed to respond only to higher spatial frequencies than the ones of the gaussian shape, is much less affected. This is due to the fact that the statistical calculation suffers from a significant contrast drop due to the difference of intensity between the center of the beam and its edges, and any contrast change additional to that one is much less effective.

Figure 4 presents the photoacoustic signal as it was recorded by the purposed setup, considering the sample and pump laser described above (black). As a reference, a simultaneous contact-based piezoelectric sensor was also recorded, and presented below (gray). The similarity between the overall behavior of the two signals is well observed.

## 4. Conclusions and Future Work

This manuscript describes the foundations, as well as experimental demonstration of an all-optical speckle contrast sensing setup. The setup is an advancement of previously developed methods using speckle contrast for various use cases; from detection of vibration in many frequency ranges, to other temporally changing scattering behaviors.

The method relies on a spectral estimation of contrast, rather than the traditional statistical one. The validity of such estimation is examined and presented, along with its benefits over the traditional calculation, especially in eliminating low frequency intensity variations shadowing speckle contrast.

Lastly, the method is demonstrated primarily on a photoacoustic signal vibration setup, continuing the original speckle contrast vibration sensing development by [13].

In addition, the method is shown to be applicable also for phonocardiography and blood-flow-related speckle changes signal. Detection of vibration of significantly different frequency band compared to the photoacoustic scenario is presented in Figure 5. In this case, the probe was pointed onto a human chest, measuring phonocardiac signals originating in heart beats (and commonly detected using a stethoscope). The same ~1Hz periodicity is observed in this case as well, in addition to the well-known double pulse (S1–S2) characterizing acoustic chest vibration signals [21].

Figure 6 describes the usage of all optical speckle contrast sensor in a non-vibration application. The speckle contrast reduction due to blood red cell scattering, resembling blood flow, is shown in this figure. A ~1 Hz periodicity can be seen in this signal, characterizing blood flow. This idea was realized in a common speckle contrast approach in [20].

## Figures and Tables

**Figure 1 sensors-22-03250-f001:**
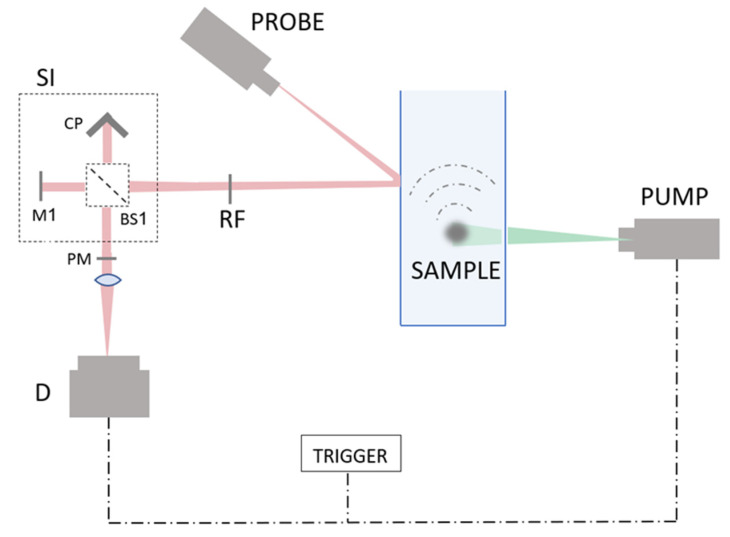
The system. The PUMP is the photoacoustic excitation laser relevant only for the photoacoustic part of the manuscript. The detection system consists of all the components on the right side of the sample. A PROBE laser illuminates the vibrating object. The reflected speckle pattern is diffused by a rotating diffuser RF, moving forward to a shearing interferometer SI, containing a corner prism CP, a mirror M1, and a beam splitter BS1. Then, the generated spectrum passes through a phase plate PM which retards central low frequency by π and thus, creates subtraction of the low frequencies from the high frequencies. The two channels are focused onto the detector D.

**Figure 2 sensors-22-03250-f002:**
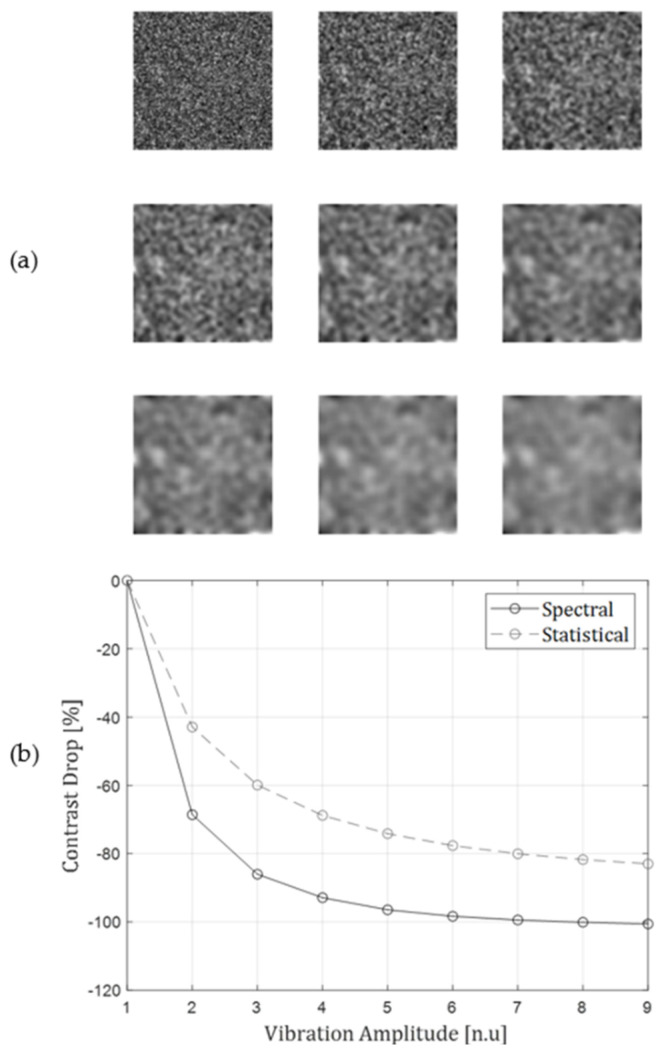
(**a**) A set of simulated speckle pattern in different contrast affecting scenarios. Each image represents a degrading contrast of the speckle pattern, due to heavier smoothing. (**b**) A comparison between contrast drop of traditional statistical calculation, and the suggested optical spectral estimation, for the corresponding images from (**a**).

**Figure 3 sensors-22-03250-f003:**
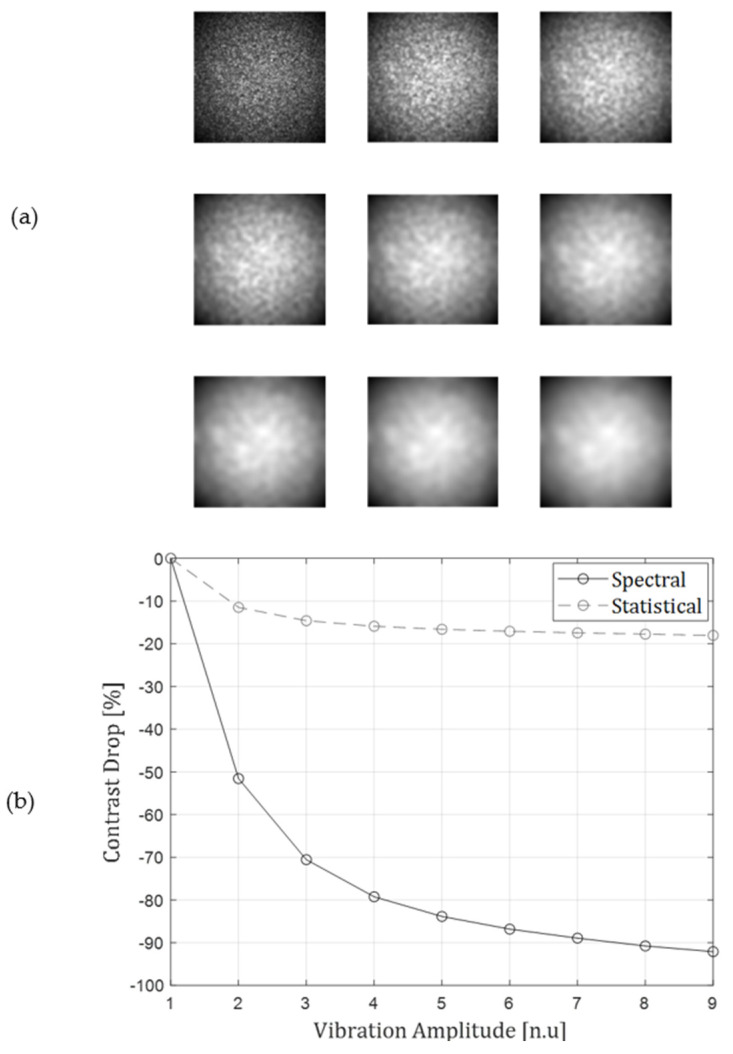
Contrast estimation method response to a non-uniform intensity distribution. (**a**) A set of speckle images similar to the ones presented in Figure 2, only this time not with uniform illumination, but a gaussian intensity distribution. (**b**) The corresponding contrast values for the set of patterns from both estimation methods.

**Figure 4 sensors-22-03250-f004:**
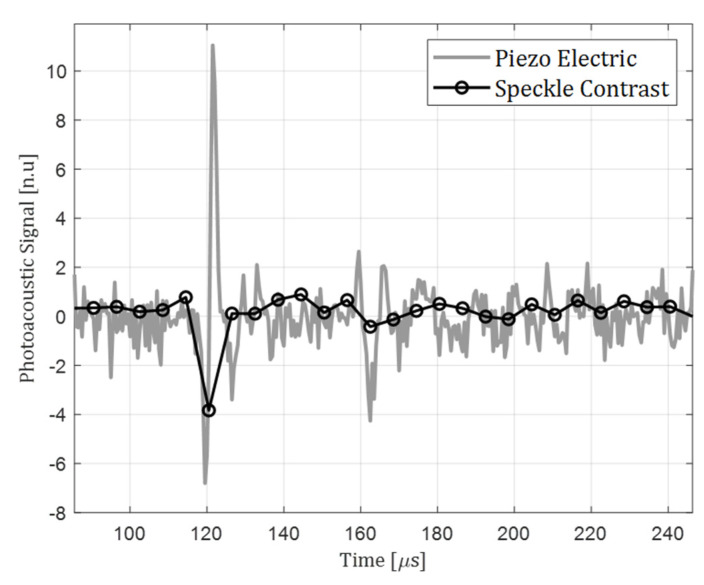
Photoacoustic signal reconstruction-compared to piezo electric transducer.

**Figure 5 sensors-22-03250-f005:**
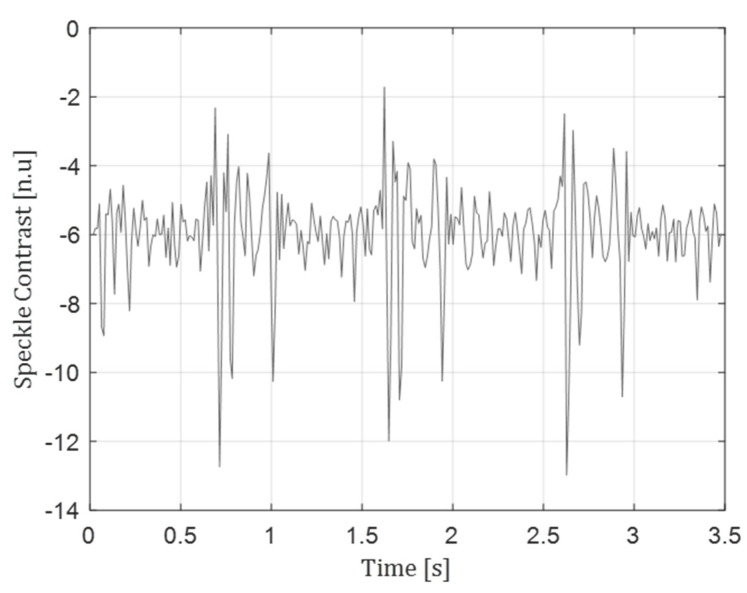
Chest movement signal reconstruction.

**Figure 6 sensors-22-03250-f006:**
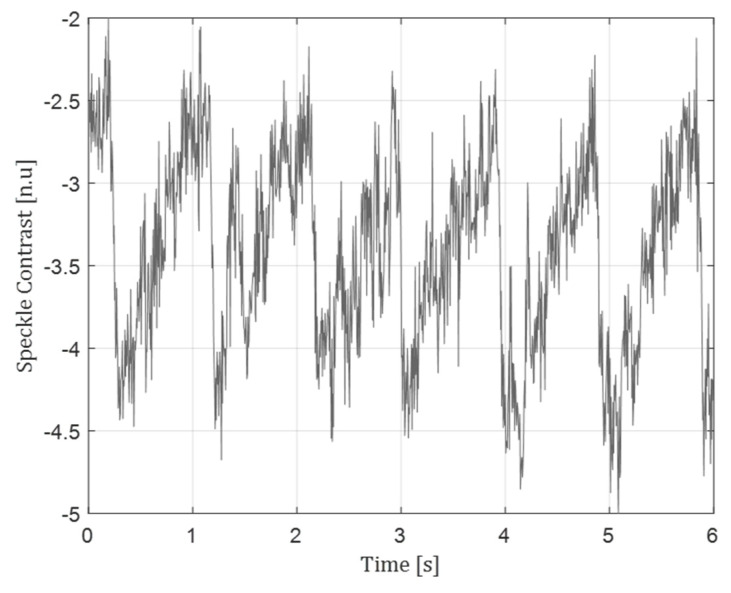
Blood flow signal reconstruction.

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
