# Peer review of "All Optical Speckle Contrast-Based Vibration Sensor for Photoacoustic Signal Detection"

_sensors, 2022, doi:10.3390/s22093250_

Round 1

Reviewer 2 Report

The manuscript, entitled: “All Optical Speckle Contrast based Vibration Sensor for Photoacoustic Signal Detection,” presented an all optical, non-contact, vibration sensor based on optical speckle contrast, and demonstrated its application for photoacoustic signal detection. It is a very interesting technique, and the presentation is solid.  Comparing with previous development from the same research group, the current investigation used a single-pixel detector instead of a camera.

  1. The abstract and the title do not match. The abstract should be re-written to emphasize more of the photoacoustic application. As written, it does not seem to have too much to do with photoacoustic detection.

  1. As an improvement in comparison with previous camera-based approach, what is the advantage? Can you compare results from both approaches?

  1. Only single-point detection was provided. How do you plan to form an image?

  1. Please quantify the SNR and resolution. Based on signals from Figure 4, the resolution is much worse than that of the PZT transducer.

  1. It is unsure what is the role of Figure 5. If you want to present the technique as a more general technique and not just for photoacoustic detection, please revise your title and introduction section of the manuscript.

Reviewer 3 Report

In this paper, the authors describe a method based on vibration sensing to all-optical speckle contrast. Comparing it to traditional contrast estimation, and showing that the method developed for them uses a spectral estimation of contrast, instead of the traditional statistical one. Moreover, they also demonstrated its applicability for phonocardiography and blood flow-related speckle changes signal.

Although I feel, the authors should review the format and grammar carefully. I recommend the acceptance of this paper after minor revision.

The author should consider these comments before its publication:

- The authors should add more information about Figure 2 in the manuscript and also in the subscription.

- There are some typos and grammatical errors. Please check the manuscript carefully.

For instance:

- page 1, line 10: replace “First, it, relies..” with “ First, it relies..”

- page 1, line 12: replace “ interferometer. the” with interferometer. The”

- page 1, lines 26-28: the letter size is different

- page 1, line 32: i.e. should be in italics

- page 2, line50: a point fault after “[11-13]”

- page 2, line 54: replace “time.more” with “time. More”

- page 3, line 98: a point fault after “ background illumination”

- page 5, lines 175-182: The format of the subscription of Figure 1 is not adequate.

- page 5, line 184: there is a “(“ but not a “)”

- page 5, line 185: replace “(THORBARS). the probe” with “(THORBARS). The probe”

- page 7, line 243: replace “Figure 6. in this…” with “Figure 6. In this..”

-page 9, line 251: replace “5. Conclusion” with “4. Conclusion”

- page 9, line 268: there is a ” at the final of the sentences but not a “ before it.

- There are some mistakes in the format of the bibliography.

For example:

- There are some without the final page (ref: 2, 5, 6, 8, 9, 12, 13, 15)

- There are some without DOI (ref: 4, 18).

Round 2

Reviewer 2 Report

I do not have further comments